# Spatiotemporal Activity Mapping for Enhanced Multi-Object Detection with Reduced Resource Utilization

**Shashank * and Indu Sreedevi**

Department of Electronics and Communication Engineering, Delhi Technological University, Delhi 110042, India
*   Correspondence: shashank.ece.dtu@gmail.com or shashank_2k17phdec05@dtu.ac.in

**Abstract:** The accuracy of data captured by sensors highly impacts the performance of a computer vision system. To derive highly accurate data, the computer vision system must be capable of identifying critical objects and activities in the field of sensors and reconfiguring the configuration space of the sensors in real time. The majority of modern reconfiguration systems rely on complex computations and thus consume lots of resources. This may not be a problem for systems with a continuous power supply, but it can be a major set-back for computer vision systems employing sensors with limited resources. Further, to develop an appropriate understanding of the scene, the computer vision system must correlate past and present events of the scene captured in the sensor's field of view (FOV). To address the abovementioned problems, this article provides a simple yet efficient framework for a sensor's reconfiguration. The framework performs a spatiotemporal evaluation of the scene to generate adaptive activity maps, based on which the sensors are reconfigured. The activity maps contain normalized values assigned to each pixel in the sensor's FOV, called normalized pixel sensitivity, which represents the impact of activities or events on each pixel in the sensor's FOV. The temporal relationship between the past and present events is developed by utilizing standard half-width Gaussian distribution. The framework further proposes a federated optical-flow-based filter to determine critical activities in the FOV. Based on the activity maps, the sensors are re-configured to align the center of the sensors to the most sensitive area (i.e., region of importance) of the field. The proposed framework is tested on multiple surveillance and sports datasets and outperforms the contemporary reconfiguration systems in terms of multi-object tracking accuracy (MOTA).

**Keywords:** computer vision; activity mapping; reconfiguration; multi-object tracking; spatiotemporal analysis; federated optical flow

## 1. Introduction

Computer vision applications have experienced tremendous growth over the past two decades. Most commonly, the computer vision applications cover the automotive, sports, entertainment, healthcare, robotics, security and surveillance areas [1]. Computer vision systems perform data processing to obtain an understanding of a scene. Systems employing robotic vision [2] take it to the next level by actuating actions based on the understanding developed by computer vision. Most of the modern computer vision applications rely on one or more sensors co-operatively sensing their environment to detect and track objects of interest and obtain information of activities or events in the surroundings.

The performance of a computer vision application depends on the accuracy of the data captured by the sensors used in terms of application-specific information. An accurate computer vision system is capable of distinguishing object(s) of interest prior to capturing the image of the scene, so the data bearing high information of the relevant object(s) of interest can be made available for processing. Further, if the object(s) of interest are captured in the center of the sensor's field of view (FOV), the information and understanding of the event obtained by processing the data are optimized. The performance of the computer

vision system relies on the processing of the data bearing information about the object(s) of interest that are captured by the sensors, whereas the sensors can be made to capture the data with high application-specific accuracy by way of calibration. Therefore, since the sensor calibration and data processing of computer vision systems are inter-related [3], the re-configuration of sensors in real time is a major challenge, especially for systems with limited resources, which cannot afford a high computational complexity. Therefore, there is a need for a framework for sensor reconfiguration that can efficiently address the challenge of limited resources.

Applications such as driverless automobiles [4], sports analytics [5] and the surveillance of large areas using UAVs [6,7] employ mobile sensors to capture the environment. The sensors of such applications often have limited resources; therefore, optimized resource utilization becomes essential. In some real-time computer vision applications, due to high computational complexity requirements, the resources are exhausted at a higher pace; thus, resources become critical.

Active computer vision systems (also known as active vision systems) are capable of calibrating (reconfiguring) the internal and external parameters of their sensors according to the needs of the system. Most of the active vision systems, such as the systems proposed in [8–10], rely on a pre-defined prioritization of the area under observation, based on assumptions regarding the sensor's field of view and surveillance area. As the activities are highly dynamic in occurrence, the pre-defined placement and orientation of the sensors result in an inappropriate sensor's pose, which hardly enables the object(s) of interest to be captured in the center of the sensor's FOV. Such a setting of the sensors in the network results in extraction of data with insufficient information of an activity or event.

It should be noted that the pattern of activities also affects the understanding of an event. A computer vision system must therefore take into consideration the events and activities of the past as well as the present activities when developing an understanding of a scene. For example, in a surveillance system, if an event occurs repetitively, the computer vision system must identify the area or site of the event and must prioritize that area or site for inspection. However, an active vision system requires highly complex computations to obtain such an understanding, utilizing a lot of resources.

Based on the abovementioned challenges, a computer vision system capable of reconfiguring the sensors based on spatiotemporal activity analysis, utilizing low resources, is desirable. This article provides a simple yet effective framework for the reconfiguration of sensors participating in an active vision system, thus yielding a better activity analysis and scene understanding with a very low computation complexity.

The framework performs a spatiotemporal evaluation of the scene to generate adaptive activity maps, based on which the sensors are reconfigured. To minimize the resource utilization, the proposed framework proposes a model-based approach (i.e., a non-learning-based approach) for the spatiotemporal evaluation of the scene by utilizing simpler concepts of image processing in combination. Based on the spatiotemporal activity analysis of the scene, the framework assigns a normalized sensitivity value to each pixel of the frame, which represents the impact of present and past activities on the corresponding pixel. Standard half-width Gaussian distribution is used to develop the temporal relationship between the past and present events. The framework further provides a federated optical-flow-based filter to identify and distinguish critical activities or events captured within the field of view of the sensors. Based on the normalized pixel sensitivity values of the adaptive activity maps, the framework reconfigures the sensors to align the center of the FOV of the sensors to the most sensitive portion of the scene, thus capturing the data with the best information in each frame.

The proposed framework can be utilized in a single-camera setting but is more suitable for a camera-pair configuration, with each node having two sensors (i.e., a primary camera and a secondary camera cooperatively operating together). In the camera-pair configuration, the primary camera includes basic image-processing capabilities and is used to generate activity maps for a scene. Based on the activity maps, the secondary

camera can be calibrated (or reconfigured) to obtain optimized data. The performance of the framework is determined in terms of multi-object tracking accuracy (MOTA). In the case of the single-camera setting, the framework is capable of determining object(s) of interest in the frame initially; however, the calibration of the camera may result in the loss of a new object of interest being captured by the sensor, as the field of view (FOV) of the sensor is adjusted in accordance with the knowledge of the object(s) of interest previously present in the frame. It must be noted that the objective of the proposed framework is not to obtain a high-performance activity tracking system; rather, the proposed framework intends to provide a balance between resource utilization and tracking accuracy and enables spatiotemporal activity analysis using simple image-processing methods.

## 2. Background

To capture accurate data for the optimized performance of the computer vision system, the system must identify the region of importance (ROI) in the FOV of the sensors and reconfigure the sensors to match the center of the ROI with the center of the FOV. There are several methods for determining the ROI, activity mapping [11] being one of the most efficient ones. Most systems proposed for ROI determination focus on the activities detected in the present frame for activity mapping and neglect the impact of past events (i.e., those events that are captured by a sensor in the past while capturing a frame for a timeframe) on the activity map. Such systems usually focus on reducing noise in the frame rather than establishing a spatiotemporal relationship between present and past events. The expression "past events" must be interpreted as the events captured by a sensor and detected by processing a frame that is in the past in relation to a current frame of the sensor. The expression "present events" must be interpreted as the events derived by the processing of the current frame captured by the sensor. For example, the past events can be events captured by the sensor through the analysis of the most recent frame captured by the sensor, whereas all the events derived by processing the frame prior to the current frame can be referred to as "past events". Thus, the activity map generated by such systems is only a filtered set of frames accumulated together.

It must also be noted that the system must be capable of identifying critical activities or events and distinguishing them from the undesirable activities captured in the scene. The ROI must only be specific to the desired activities dedicated to the overall functionality of the system. For example, a traffic monitoring system should avoid the falling of leaves from trees in the scene and must treat such an activity as undesirable. On the other hand, a computer vision system designed to estimate the effects of climatic change on trees must consider the falling of leaves as a critical activity and must treat other activities in the scene as undesirable.

Pan et al. [12] and Mehboob et al. [13] proposed region of interest (ROI) estimation for vehicle flow detection using morphological operations to filter noise. Pan et al. [12] proposed a self-adaptive window-based traffic estimation using background subtraction, edge detection for object detection and morphological features for noise reduction. Mehboob et al. [13] proposed centroid detection using morphological close and erode operation for noise reduction and motion vectors for traffic flow estimation. The morphological operations used in [12,13] attempted to reduce the noise but failed to prevent the detection of undesirable activities in the scene (i.e., failed to determine critical activities and prevent the detection of undesirable activities from the scene). Both systems further developed an understanding of the ROI based only on the activities in the present frame and neglected the impact of past events on the ROI. To address this gap, the approach proposed in [14] considered spatiotemporal evaluation for activity mapping. The approach proposed a spatiotemporal relationship between the present and past activities in the frame; however, it did not provide a filter to remove noise or undesirable object detection from the scene.

Contemporary approaches for activity mapping either employ highly complex computation models or artificial intelligence approaches for ROI detection. Marvin and Moritz [15] presented a non-parametric model for spatiotemporal activity based on Gaussian process

regression (GPR). Sattar et al. [16] proposed a convolution neural network (CNN)-based spatiotemporal activity mapping method for group activity detection. Zhao and Gao [17] proposed an online feature learning model for spatiotemporal event forecasting. Liu and Jing [18] proposed an artificial intelligence (AI)-based activity mapping method for sports analytics using spatiotemporal activity patterns. Yan et al. [19] proposed an end-to-end position-aware spatiotemporal activity analysis using a long short-term memory (LSTM) approach.

The approaches in [12–14] provide simple models for activity mapping and ROI detection; however, they lack the accuracy of ROI prediction. The methods proposed in [15–19] provide efficient spatiotemporal activity mapping for ROI detection, but at the cost of high computational complexity, and are thus not suitable for systems with low or limited resources. Artificial intelligence (AI)-based systems [16–19] require high computation and storage capabilities to train the model and further require iteratively changing the training model in unforeseen conditions, illumination changes, etc. AI-based multi-object tracking systems are also susceptible to adversarial attacks, which makes the timely training of the system critically necessary. The AI-based systems, due to re-iterative training to deal with unforeseen conditions and adversarial attacks, require higher computational resources. There is a trade-off between the accuracy and computational complexity of the activity mapping approaches. The proposed framework strikes a balance between resource utilization and activity tracking accuracy, enabling spatiotemporal activity analysis using simple image-processing techniques for accurate ROI detection.

## 3. Spatiotemporal Activity Mapping Model

Considering the requirement of spatiotemporal activity mapping for accurate ROI detection and better tracking accuracy, this article provides an efficient framework with a low resource utilization for computer vision systems with limited resources. The framework strikes a balance between resource utilization and activity tracking accuracy, allowing for spatiotemporal activity analysis using simple image-processing techniques. The framework (as shown in Figure 1) generates adaptive spatiotemporal activity maps based on the sensed data obtained by the primary camera in consecutive temporal frames, which can be processed to derive information to calibrate the secondary camera of the computer vision system. Each adaptive spatiotemporal activity map is derived by the past frames and the present frame for each timeframe. The importance of the past frames and the present frame is derived through a normalized half-width Gaussian distribution function such that events in the present frame are assigned the highest importance, whereas the events in the past frames are assigned importance across the normalized half-width of the Gaussian distribution function according to their relative timeframes. The adaptive activity map is further used to determine the dynamic ROI of the scene. The framework further includes a re-configurator that alters the configuration space of the sensor such that the center of the ROI matches with the center of the sensor's FOV. The revised reconfiguration results in the improved accuracy of activity tracking and thus yields better scene understanding.

### *Notations Used:*
**$k$:** Number of past frames captured by the primary camera;
**S(t):** Data sensed by the primary camera in the present frame;
**S(t − i):** Data sensed by the primary camera in "i" frames prior to the present frame;
**$S_{1i}$:** Data obtained by object detection on $S(t − i)$;
**$S_{2i}$:** Data obtained by applying adaptive thresholding on $S_{1i}$;
**$S_{3i}$:** Data obtained by the binarization of $S_{2i}$;
**$S_{4i}$:** Data obtained by filtering $S_{3i}$ using federated optical flow;
**H(t − i):** Temporal function for "i" frames prior to the present frame;
**$S_{5i}$:** Temporal component of $S_{4i}$ obtained as a product of $S_{4i}$ and $H(t − i)$;
**X:** Cumulative spatiotemporal activity map for the present frame;
**N:** Normalized spatiotemporal activity map for the present frame;
**R:** Reconfiguration parameters for the secondary camera;



**C:** Data from the calibrated secondary camera; and
**Y:** Activity analysis after processing C.

The following subsections of Section 3 intend to discuss the functionality of the proposed framework for spatiotemporal activity mapping. Through Section 3.1, we discuss some basic image-processing tools/methods used by the proposed framework for generating adaptive spatiotemporal activity maps, whereas the steps are discussed in Section 3.2.

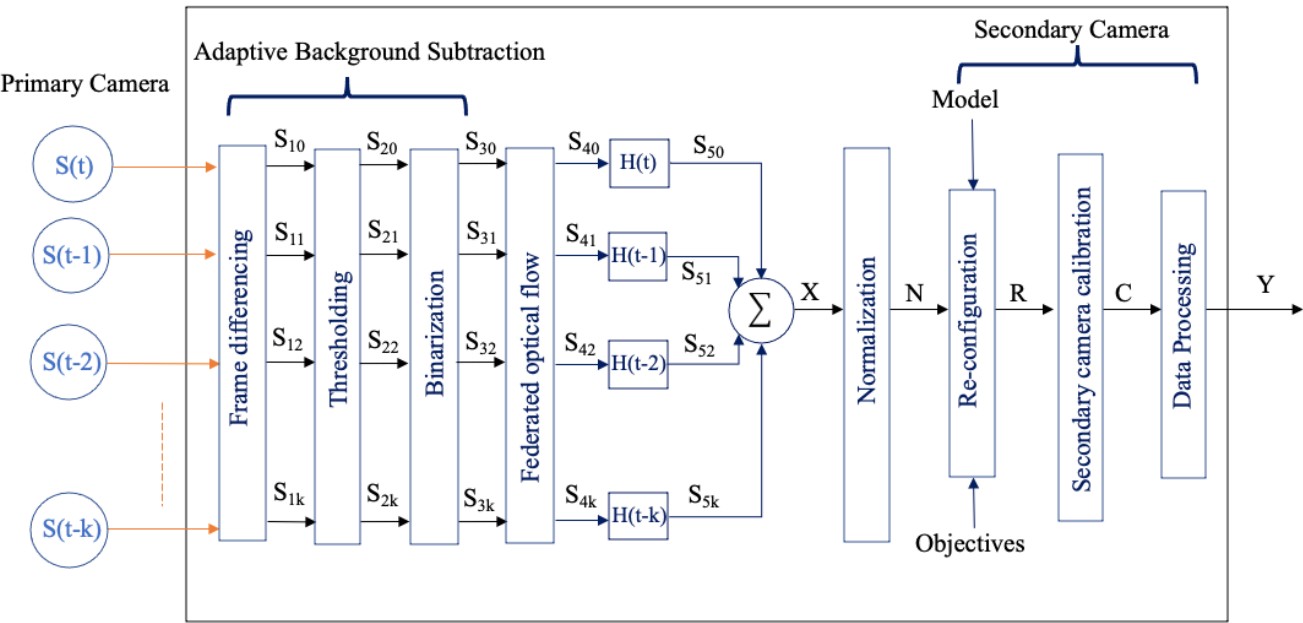

**Figure 1.** Adaptive spatiotemporal activity mapping framework.

*3.1. Methods*

- *Adaptive background subtraction*

To detect the moving pixels in the FOV from the data captured by the sensor, the framework utilizes frame differentiation-inspired adaptive background subtraction, which enables activity region detection and background allocation for each frame. The pixel information from each preceding frame is utilized to obtain background and foreground pixels in the next frame. For example, if S(t) is the sensed data in the frame at time "t" and S(t − 1) is the sensed data in the frame at time "t − 1", then the foreground is initially extracted using frame differentiation such that each frame has its own background and foreground with reference to a relative previous frame, which can be utilized for the detection of objects in the camera FOV. The foreground information at time t (i.e., F(t)) is obtained by using Equation (1):

$$F(t) = S(t) - S(t - 1); \tag{1}$$

The foreground is further filtered (pre-processed) by adaptive thresholding followed by binarization to reduce noise from the foreground. A combination of frame differencing with adaptive thresholding and binarization results in adaptive background subtraction.

- *Normalized Half-width Gaussian Distribution*

The proposed framework uses the half-width of the normalized Gaussian distribution function to allocate temporal importance to past frames captured by the sensors. The half-width Gaussian distribution provides an accurate prioritization and importance to the past events captured by the sensor. Due to a continuous curve, the half-width Gaussian distribution provides flexibility such that it can be fragmented into any number of segments and thus is the most suitable function for the temporal relationship between the present and past frames captured by the sensor. Further, as the Gaussian distribution finds vast

applications in building temporal relationships [20,21], thus, to relate the past events to the present activities, the half-width of the normalized Gaussian distribution has been used. For the temporal relationship between the past and present frames, we used the first half of the standard normalized Gaussian distribution curve (as shown in Figure 2).

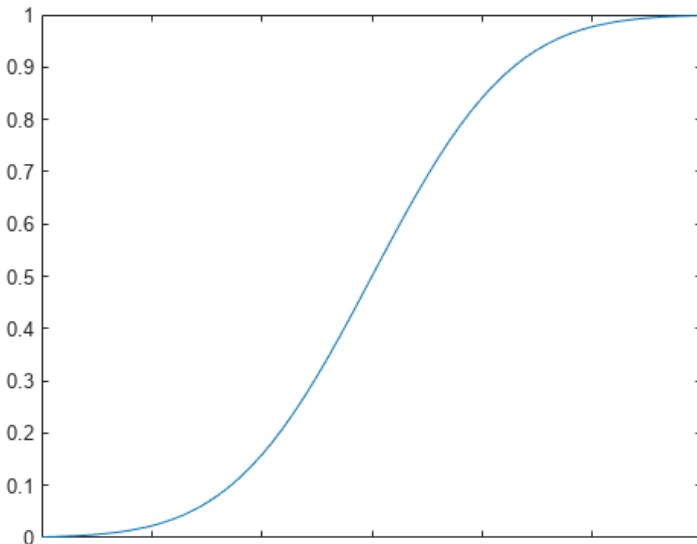

**Figure 2.** Half-width Gaussian Distribution function.

The normalized half Gaussian distribution function is used to assign importance to events in the present and past frames in order to assign a spatiotemporal relationship between consecutive frames captured by the primary camera. Specifically, the present events are assigned a normalized importance "1", whereas the past events are assigned a normalized importance value according to the normalized half-width Gaussian curve and the frames relative to the time of the present frame. The half-width Gaussian curve is a continuous curve, can be segregated into any number of values temporally and is thus ideal for the spatiotemporal relationship of events.

- *Federated Optical flow*

Federated systems [22,23] combine information from various sources to reach a consensus decision. The adaptive behavior of federated systems enhances the system's overall performance. Optical flow methods [24–26] have been extensively used to track the motion sequence of a group of pixels in consecutive frames. The two assumptions for obtaining optical flow are: (i) the group of pixels moves simultaneously in the consecutive frames and (ii) the illumination does not change in consecutive frames. In [27], Iqbal et al. presented the use of optical flow and Lukas–Kanade approaches in computer vision applications and activity mapping. The movement of pixels is obtained by determining the change in position of a pixel or cluster of pixels relative to the neighbor pixels. In the proposed framework, federated optical flow is used to design a filter to avoid the impact of undesirable activities on the ROI. The proposed framework determines the optical flow in each consecutive temporal frame captured by the sensor. Further, to filter out the unwanted objects or noise from the pre-processed foreground of each frame, the pixels from each temporal frame obtained after adaptive background subtraction are segregated into clusters by determining clusters of pixels with a consistent optical flow in consecutive temporal frames. Adaptive thresholding is used on the obtained optical flow to enhance the accuracy of the segmentation of the ROI, thus resulting in a federated optical flow.

*3.2. Process*

The proposed framework is designed to detect important activities captured over a period of time in the sensor's FOV and generate an adaptive activity map for the scene.

Each pixel in the adaptive activity map is assigned dynamic pixel-sensitivity values to determine regions of importance in the scene. The adaptive activity map is further used to reconfigure the configuration space of the sensor such that the orientation of the sensor is altered to capture the ROI in the center of the sensor's FOV. A process flow of the framework is shown in Figure 3.

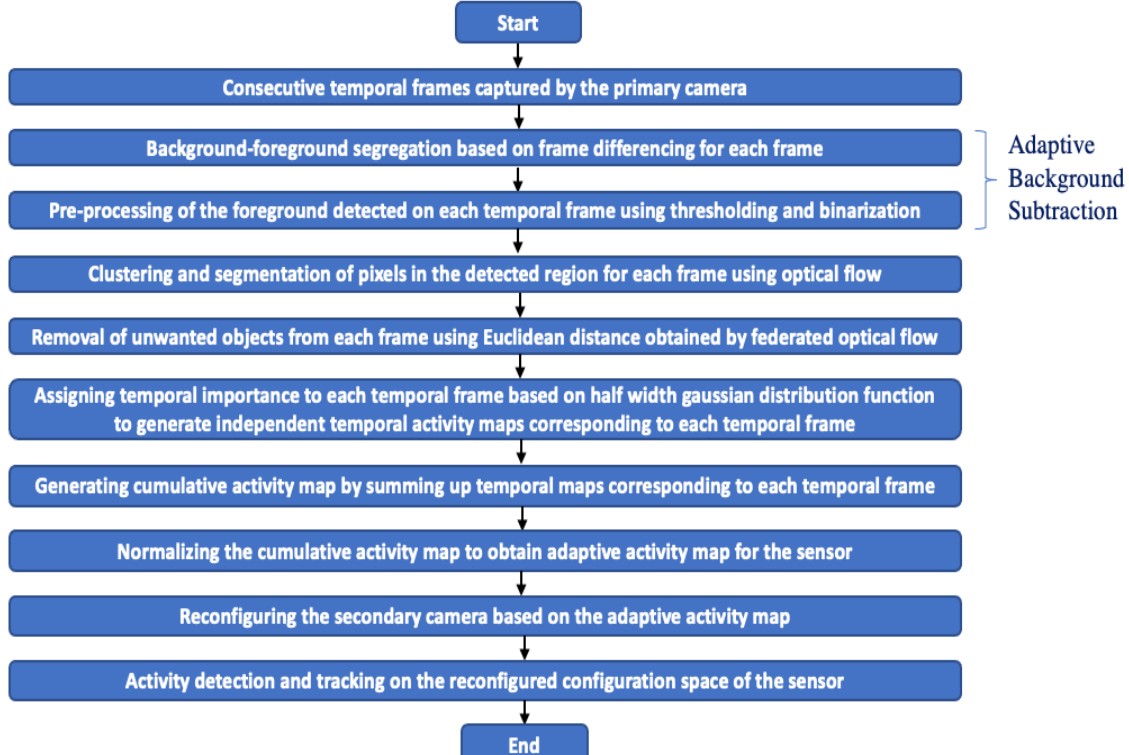

**Figure 3.** Process flow for adaptive spatiotemporal activity mapping.

Initially, the primary camera is configured to obtain data over time in the form of temporal frames. Each temporal frame is assigned a unique reference number based on the timestamp by the primary camera for relative reference. The data acquired by the primary camera in each temporal frame are processed to detect the foreground and background using adaptive background subtraction. The proposed framework, by way of frame differencing, generates one or more initial or unfiltered regions of importance (ROI) for each temporal frame from the determined foreground. The initial ROI for each temporal frame is pre-processed through adaptive thresholding, followed by binarization to filter the initial ROI in order to reduce noise from the foreground. A combination of frame differencing with adaptive thresholding and binarization results in adaptive background subtraction.

After pre-processing, the initial ROI of each temporal frame is clustered and segmented using optical flow. Undesirable objects are removed from each temporal frame using thresholding on a Euclidean distance obtained through federated optical flow. The filtered regions of importance of each temporal frame are assigned a temporal importance based on their relative temporal positions and half-width Gaussian distribution function, generating independent temporal maps corresponding to each temporal frame. The framework further proposes cumulating the independent temporal maps and normalizing the cumulative temporal map to generate the adaptive activity map for the scene. The regions of importance in the adaptive activity map are determined on the basis of normalized pixel sensitivity values (which are assigned to each pixel in the activity map), thus providing a pixel-wide, accurate spatiotemporal activity mapping of the scene.

The framework further utilizes a re-configurator for the reconfiguration of the secondary camera's configuration space such that the center of the ROI can be captured in the center of the FOV of the secondary camera. The re-configurator is provided with an objective specific reconfiguration model. The objectives further decide the overall functionality of the system and thus decide which activities are critical or undesirable. The adaptive activity maps are used by the re-configurator system to alter the orientation of the secondary camera and capture data with higher accuracy, thus providing better activity detection, tracking accuracy and scene understanding. In a single-camera setting, the primary camera acts as the secondary camera, and upon the generation of the adaptive spatiotemporal activity maps, it is configured to be calibrated based on the spatiotemporal activity map corresponding to the current (or latest) temporal frame.

Our contribution through this framework is twofold. Firstly, the present framework provides a spatiotemporal relationship between past and present events (or frames) captured by a sensor (i.e., the primary camera) to generate adaptive spatiotemporal activity maps for each frame, with the impact of past and present events on each activity map. The impact can be quantified in terms of a normalized sensitivity value assigned to each pixel of the activity map. Secondly, to remove the noise and improve the accuracy of the detection of objects of interest in each frame, the framework proposes federated optical-flow-based filtering prior to generating the adaptive spatiotemporal activity maps such that the accuracy of the adaptive spatiotemporal activity maps is enhanced. The adaptive spatiotemporal activity maps generated by the processing data sensed by the primary camera are further utilized for the calibration of the secondary camera to capture objects of interest with a high resolution, thus resulting in data with high information. The data captured by the secondary camera further result in better object detection and tracking and thus enhanced image understanding.

The performance of the proposed framework is evaluated in terms of multi-object tracking accuracy (MOTA) by the secondary camera in terms of truly (or accurately) detected positive and negative values of pixels in each frame and falsely (or inaccurately) detected pixels in each frame. The MOTA (%) increases if the truly detected pixel count increases and the falsely detected pixel count decreases. The performance parameters are elaborated on in Section 4.

## 4. Simulations and Results

The proposed framework is tested on several datasets to evaluate the efficacy on spatiotemporal activity mapping, activity detection and tracking and scene understanding. For illustration, we tested the framework using a single-camera setting on several surveillance and sports video datasets (10 seconds @ 30 fps each, i.e., 300 consecutive frames of $360 \times 640$ resolution for each video dataset) to obtain a spatiotemporal activity map after ten seconds for each video dataset for multi-object detection and tracking. The framework can be used for the calibration of reconfiguration parameters based on the generated activity map. However, the demonstration of the same requires primary and secondary sensors performing in real time. For illustration, the performance of the proposed framework is measured in terms of MOTA (%). Further, to demonstrate the results, the half-width of the half-maxima Gaussian distribution has been used, with the normalized values of the mean ($\mu$) and standard deviation ($\alpha$) being 1 and 0.5, respectively, for the temporal relation between the consecutive temporal frames. It must be noted that the half-width Gaussian distribution can be utilized for longer durations of activity analysis and thus should not be considered as a limitation to the framework. The half-width half-maxima (HWHM) Gaussian is represented in terms of the standard deviation ($\alpha$) by Equation (2):

$$\text{HWHM} = \sqrt{2\ln(2)}\,\alpha = 1.1799\alpha; \tag{2}$$

*4.1. Performance Parameters*

We have utilized markers on the consecutive temporal frames to obtain the true data of each temporal frame in the video dataset for the evaluation of the performance. The performance of the framework is analyzed in terms of multi-object tracking accuracy (MOTA). To determine the MOTA, the true positive pixel count (TPC), true positive pixel detection rate (TPR), false positive pixel count (FPC), false positive pixel detection rate (FPR), true negative pixel count (TNC), true negative pixel detection rate (TNR), false negative pixel count (FNC) and false negative pixel detection rate (FNR) are used as primary performance parameters. The expression for MOTA is formulated by Equation (3):

$$\text{MOTA (\%)} = \{(P_t - P_f)/P_t\} * 100; \tag{3}$$

where $P_t$ represents the total pixel count in the activity map, and $P_f$ represents the count of falsely detected or non-detected pixels in the activity map.

The total pixel count ($P_t$) in the activity map is represented by Equation (4):

$$P_t = TPC + FPC + TNC + FNC; \tag{4}$$

The count of falsely detected or non-detected pixels in the activity map ($P_f$) is represented by Equation (5):

$$P_f = FPC + FNC; \tag{5}$$

The pre-processed activity data ($A_{ij}$) provide the pixel activity value of each temporal frame. The pixel sensitivity is obtained by the cumulative weighted sum of the activity values obtained from all past and present frames. A weight ($W_k$) is assigned to each temporal frame in accordance with the HWHM Gaussian distribution. The pixel sensitivity value of the activity map is formulated by Equation (6):

$$S_{ij} = \sum_{(k)} W_k * A_{ijk} \tag{6}$$

where $k$ represents the number of frames captured by the sensor, $W_k$ represents the weight assigned to each temporal frame and $A_{ijk}$ represents the pre-processed activity data of the kth temporal frame.

Further, the normalized pixel sensitivity of each pixel in the adaptive activity map can be formulated by Equation (7):

$$S_{ij} \text{ (normalized)} = S_{ij}/Max(S_{ij}); \tag{7}$$

where $Max(S_{ij})$ is the maximum pixel sensitivity value in the activity map.

Spatiotemporal activity mapping is carried out by assigning a normalized pixel sensitivity value to each pixel in the scene. ROIs can be depicted as the clusters of pixels with a high value of normalized pixel sensitivity.

*4.2. Simulations*

We have compared our approach with the approaches proposed in [12–14] for multi-object tracking and traffic flow estimation. The simulations and results are derived using the MatLab Image Processing Toolbox on a work station (GPU) with 128 GB of random-access memory and an Intel(R) Xeon(R) Silver 4214 CPU @ 2.19–2.20 GHz. The results on randomly selected frames from the video dataset 1 after frame extraction, binarized adapted background subtraction and Lukas–Kanade optical flow estimation are shown in Figure 4. A comparison of the activity map and normalized pixel sensitivity derived from different approaches for video dataset 1 is shown in Figure 5. The results on randomly selected frames from the video dataset 2 after frame extraction, binarized adapted background subtraction and Lukas–Kanade optical flow estimation are shown in Figure 6. A comparison of the activity map and normalized pixel sensitivity derived from different approaches for video dataset 2 is shown in Figure 7. The results on randomly selected frames from

the video dataset 3 after frame extraction, binarized adapted background subtraction and Lukas–Kanade optical flow estimation are shown in Figure 8. A comparison of the activity map and normalized pixel sensitivity derived from different approaches for video dataset 3 is shown in Figure 9. The results on randomly selected frames from the video dataset 4 after frame extraction, binarized adapted background subtraction and Lukas–Kanade optical flow estimation are shown in Figure 10. A comparison of the activity map and normalized pixel sensitivity derived from different approaches for video dataset 4 is shown in Figure 11. The results on randomly selected frames from the video dataset 5 after frame extraction, binarized adapted background subtraction and Lukas–Kanade optical flow estimation are shown in Figure 12. A comparison of the activity map and normalized pixel sensitivity derived from different approaches for video dataset 5 is shown in Figure 13. The results on randomly selected frames from the video dataset 6 after frame extraction, binarized adapted background subtraction and Lukas–Kanade optical flow estimation are shown in Figure 14. A comparison of the activity map and normalized pixel sensitivity derived from different approaches for video dataset 6 is shown in Figure 15.

**A.    Video dataset 1 (Traffic surveillance):**

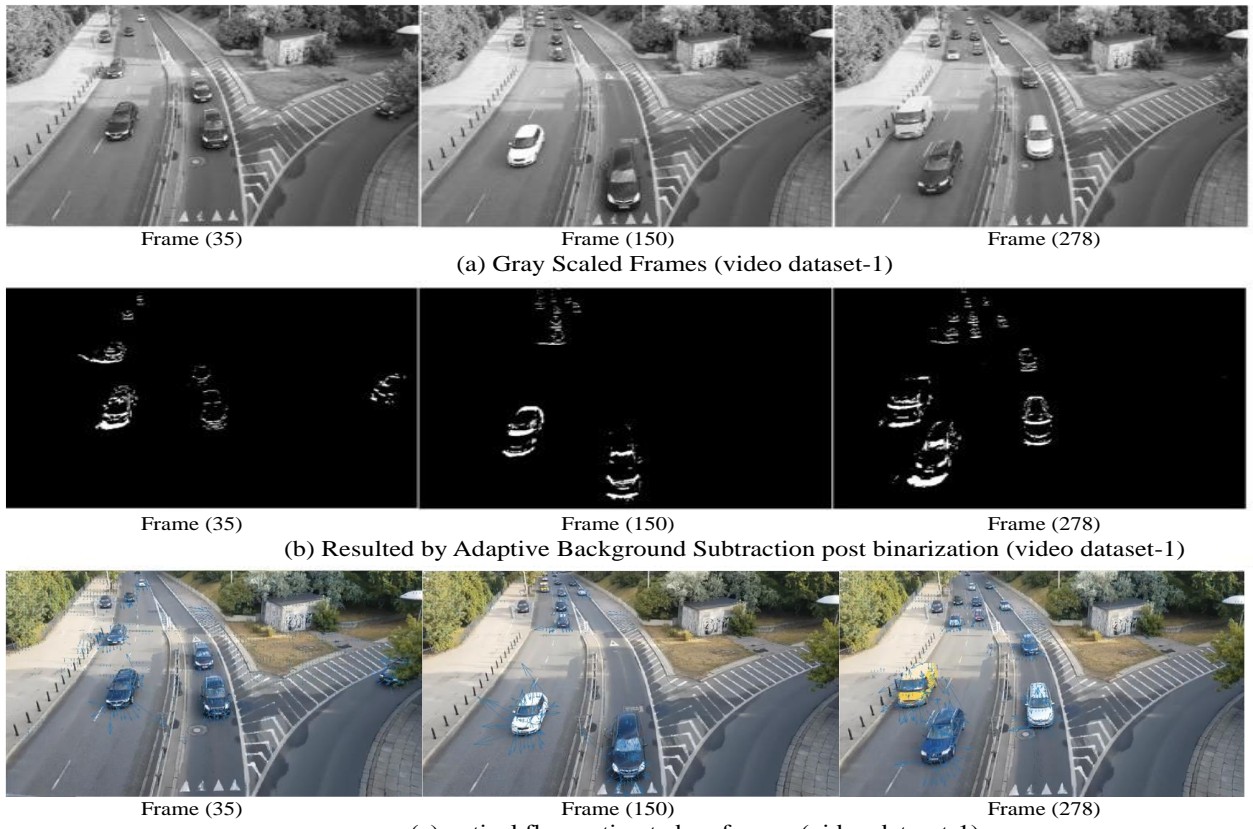

**Figure 4.** Simulation from video dataset 1 by the proposed framework.

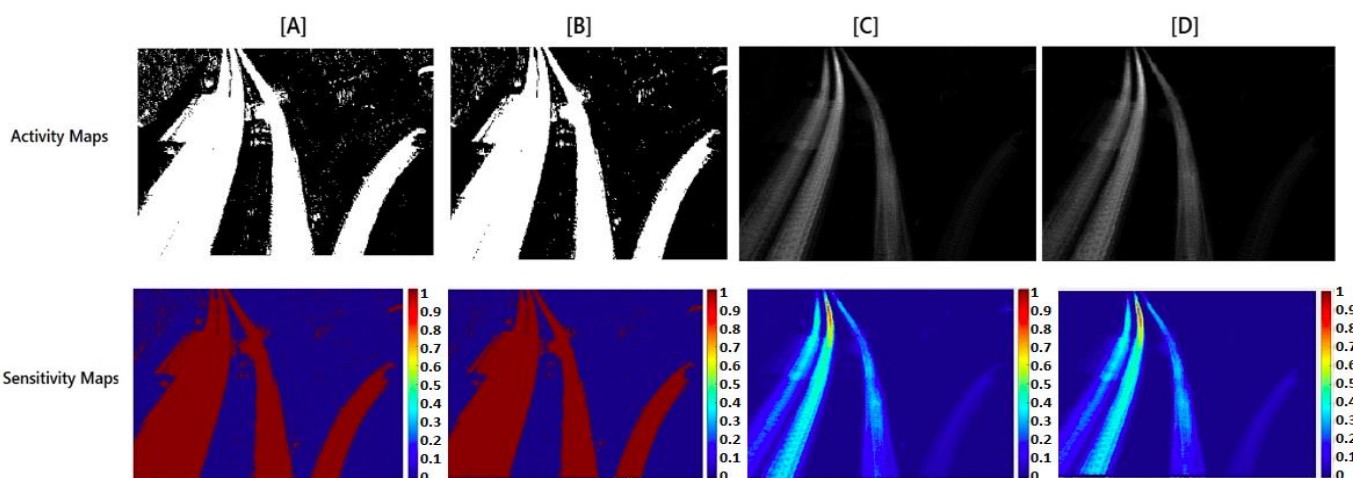

**Figure 5.** Activity map and pixel sensitivity maps of video dataset 1 by different approaches: (**A**) by Pan et al. in [12]; (**B**) by Mehboob et al. in [13]; (**C**) by Indu, S. in [14]; (**D**) our proposed framework.

**B.    Video dataset 2 (Traffic surveillance):**

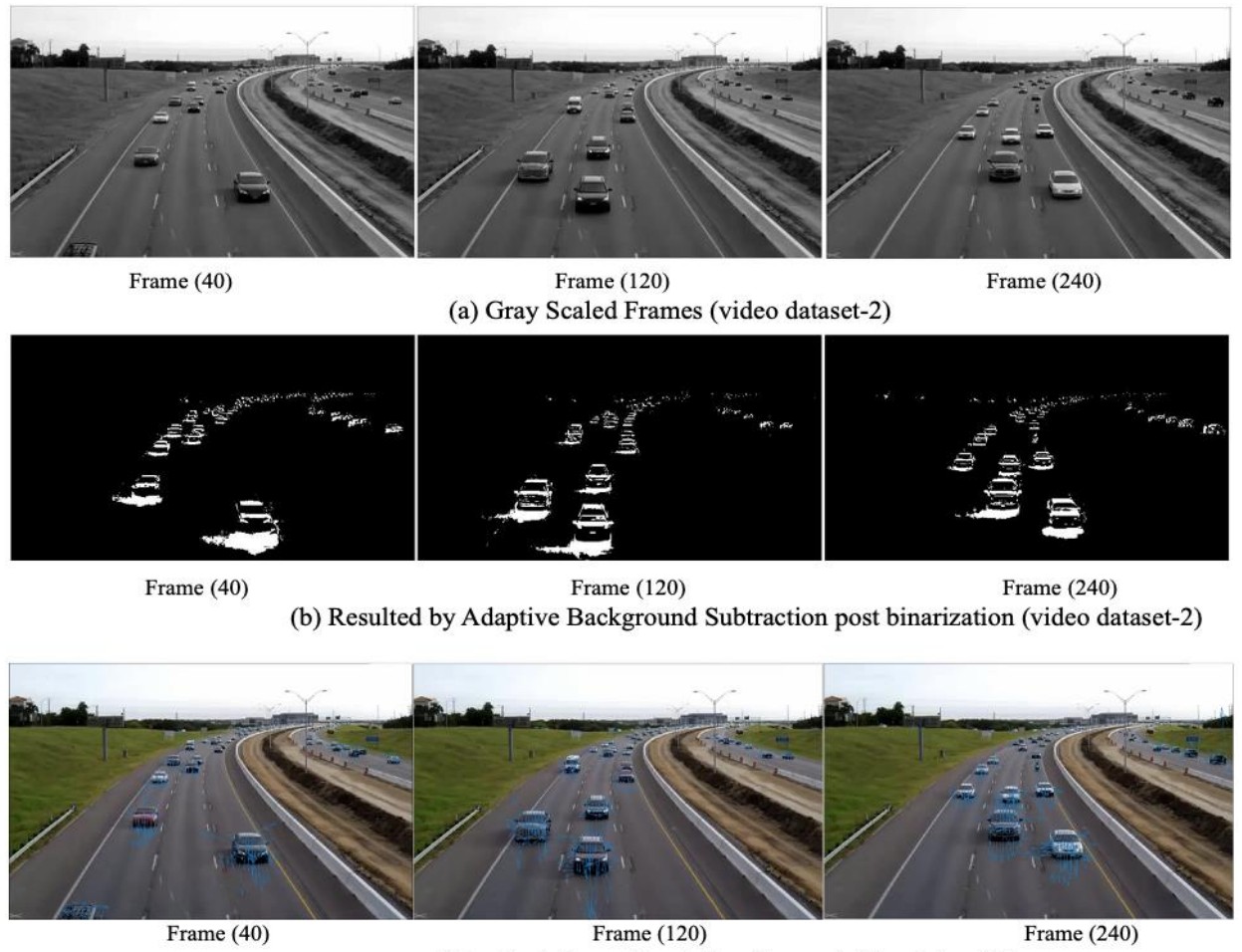

**Figure 6.** Simulation from video dataset 2 by the proposed framework.

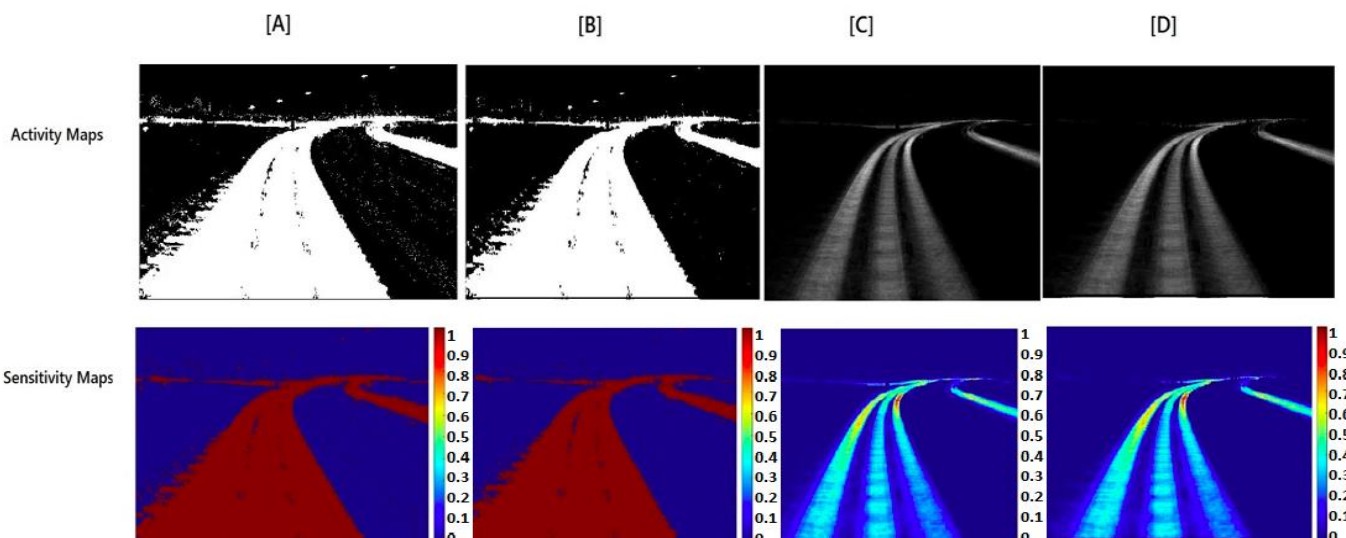

**Figure 7.** Activity map and pixel sensitivity maps of video dataset 2 by different approaches: (**A**) by Pan et al. in [12]; (**B**) by Mehboob et al. in [13]; (**C**) by Indu, S. in [14]; (**D**) our proposed framework.

### C. Video dataset 3 (Traffic surveillance):

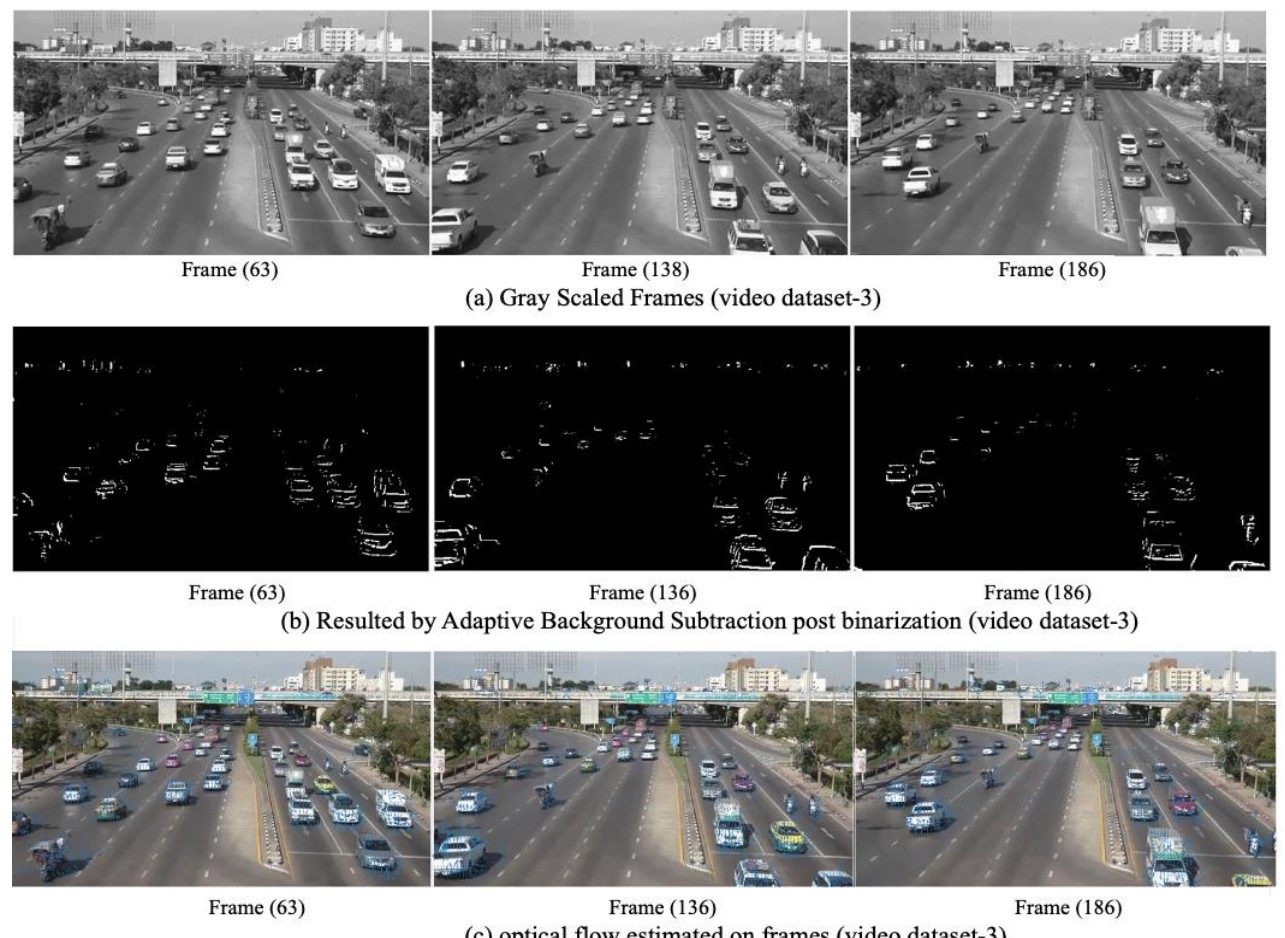

**Figure 8.** Simulation results from video dataset 3 by the proposed framework.

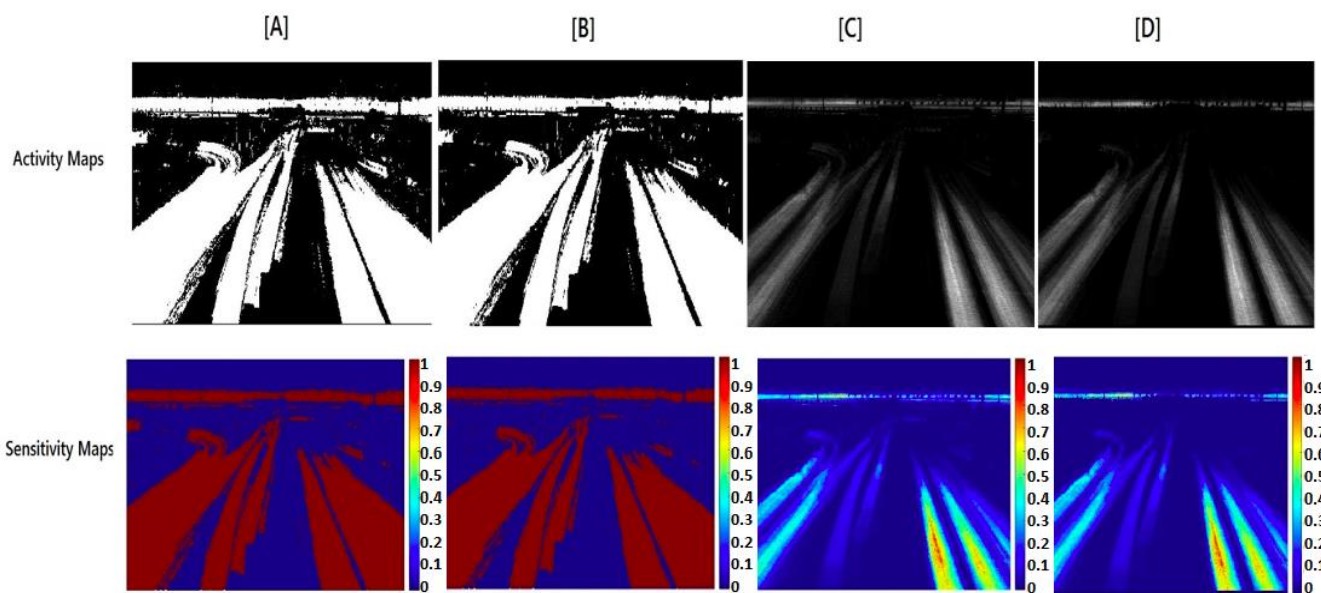

**Figure 9.** Activity map and pixel sensitivity maps of video dataset 3 by different approaches: (**A**) by Pan et al. in [12]; (**B**) by Mehboob et al. in [13]; (**C**) by Indu, S. in [14]; (**D**) our proposed framework.

### D.　Video dataset 4 (Sports—Badminton):

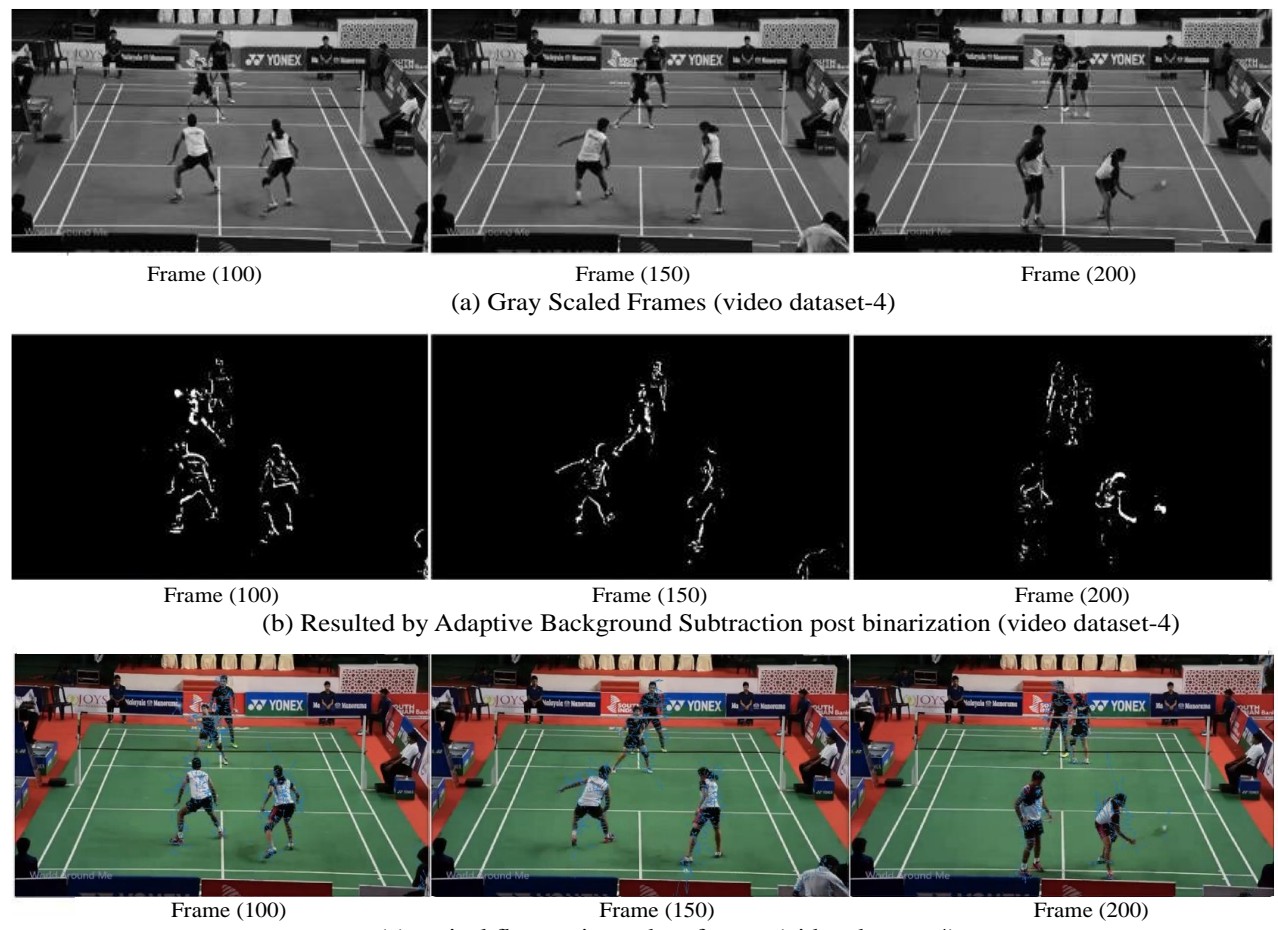

Frame (100)　　　　　Frame (150)　　　　　Frame (200)
(a) Gray Scaled Frames (video dataset-4)

Frame (100)　　　　　Frame (150)　　　　　Frame (200)
(b) Resulted by Adaptive Background Subtraction post binarization (video dataset-4)

Frame (100)　　　　　Frame (150)　　　　　Frame (200)
(c) optical flow estimated on frames (video dataset-4)

**Figure 10.** Simulation results from video dataset 4 by the proposed framework.

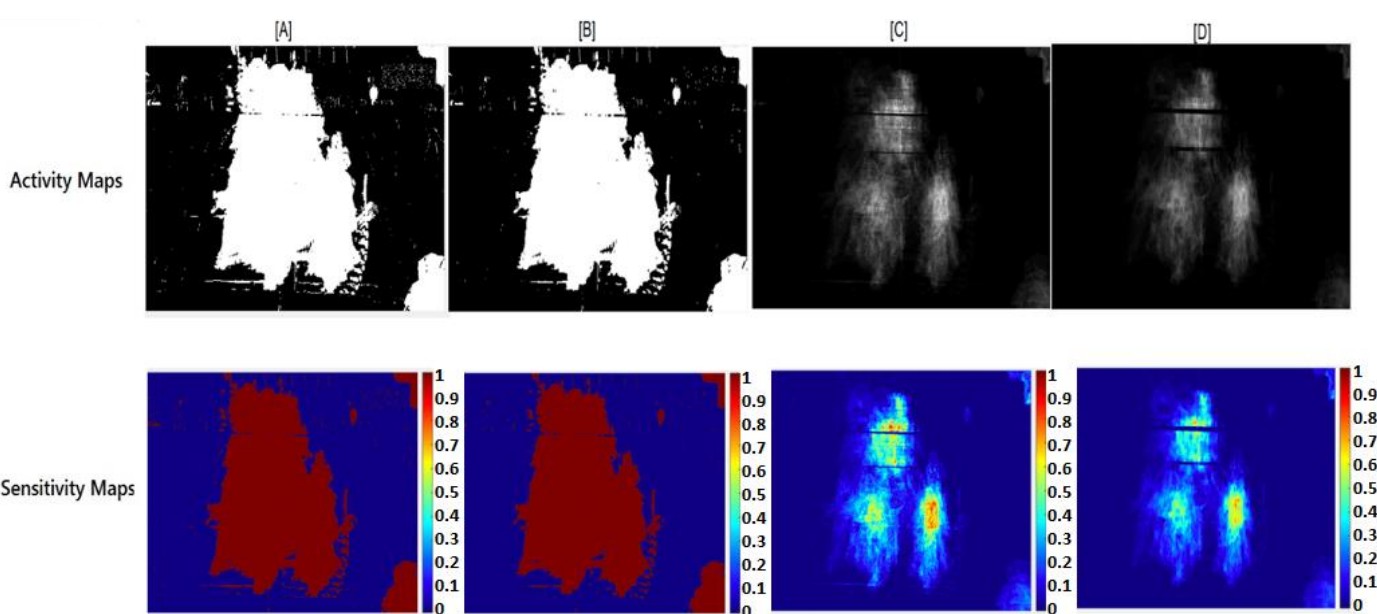

**Figure 11.** Activity map and pixel sensitivity maps of video dataset 4 by different approaches: (**A**) by Pan et al. in [12]; (**B**) by Mehboob et al. in [13]; (**C**) by Indu, S. in [14]; (**D**) our proposed framework.

### E.    Video dataset 5 (Sports—Sword fight):

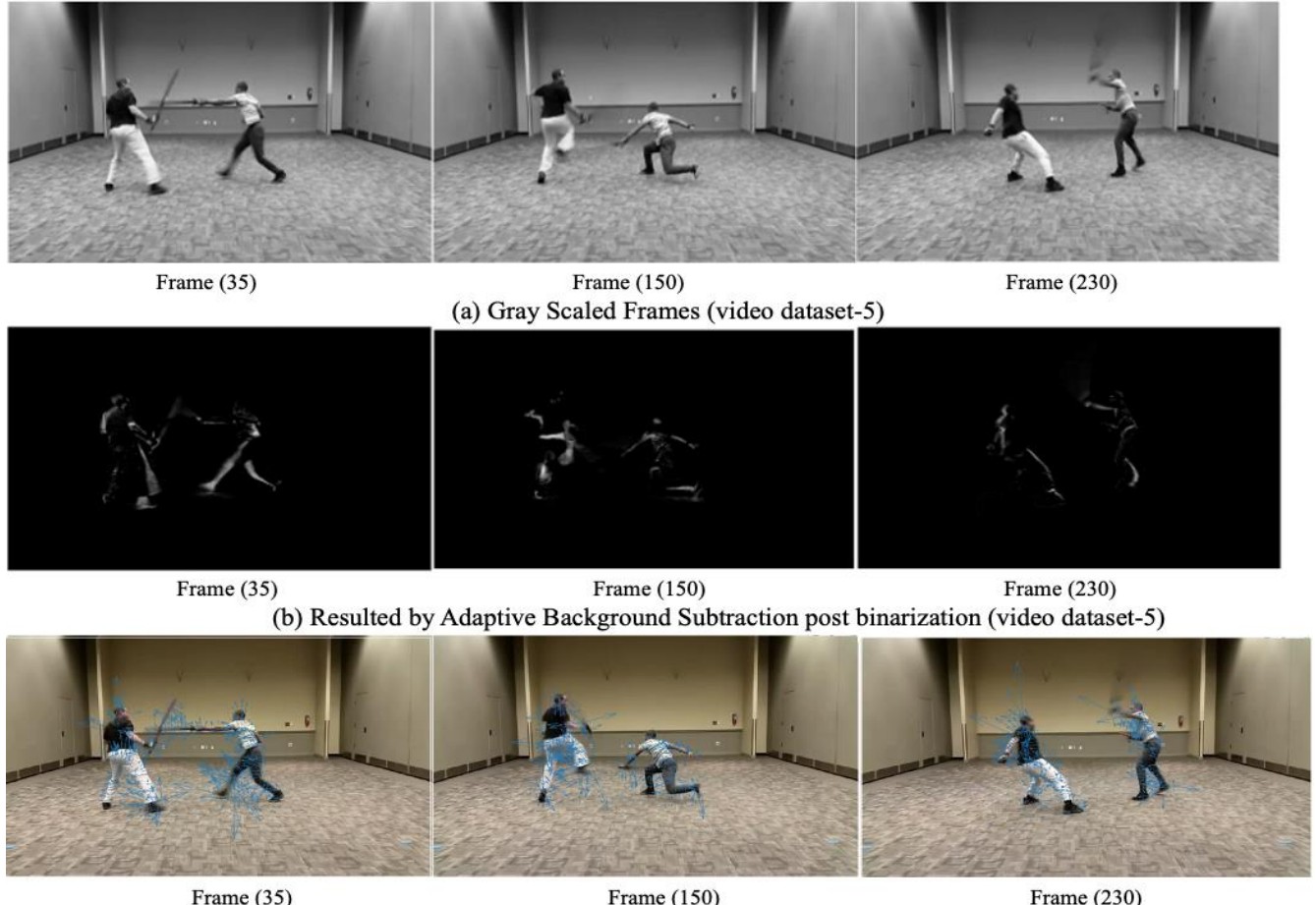

**Figure 12.** Simulation results from video dataset 5 by the proposed framework.

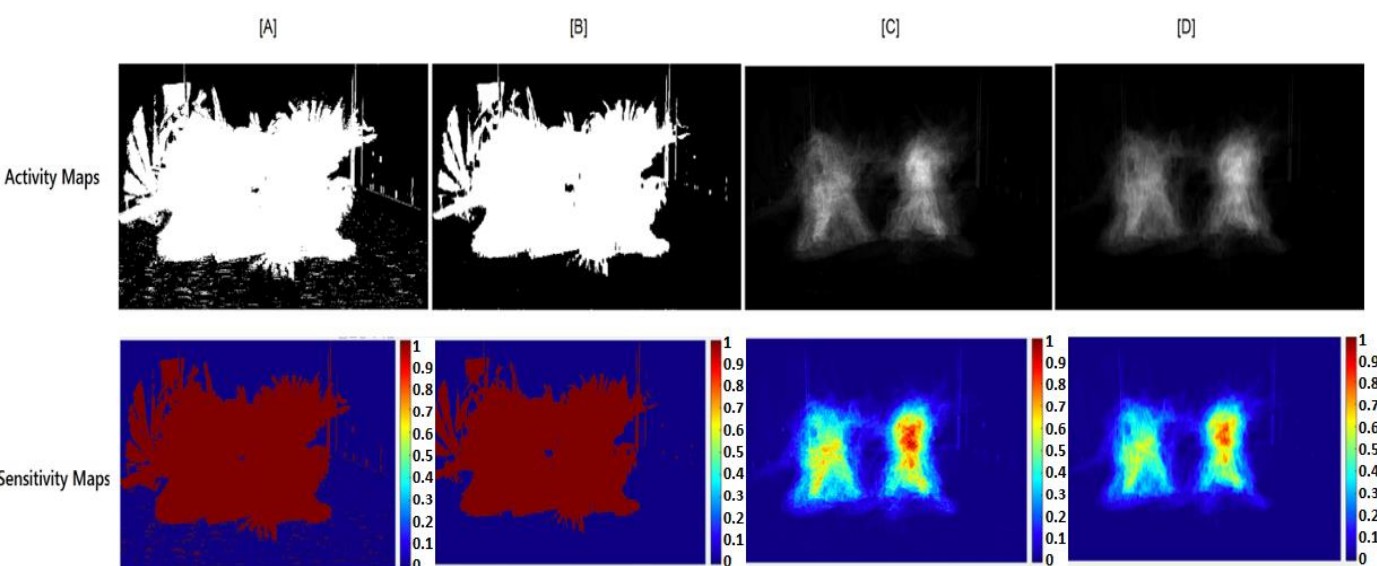

**Figure 13.** Activity map and pixel sensitivity maps of video dataset 5 by different approaches: (**A**) by Pan et al. in [12]; (**B**) by Mehboob et al. in [13]; (**C**) by Indu, S. in [14]; (**D**) our proposed framework.

**F.     Video dataset 6 (Sports—Tennis):**

Frame (50)                Frame (150)                Frame (200)
(a) Gray Scaled Frames (video dataset-6)

Frame (50)                Frame (150)                Frame (200)
(b) Resulted by Adaptive Background Subtraction post binarization (video dataset-6)

Frame (50)                Frame (150)                Frame (200)
(c) optical flow estimated on frames (video dataset-6)

**Figure 14.** Simulation results from video dataset 6 by the proposed framework.

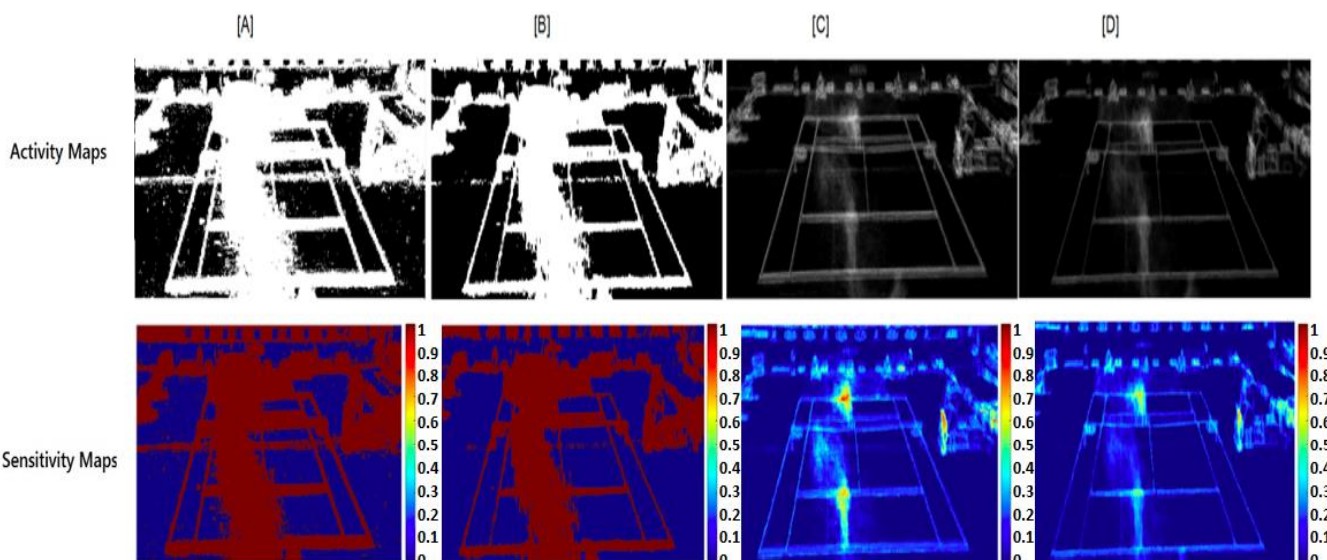

**Figure 15.** Activity map and pixel sensitivity maps of video dataset 6 by different approaches: (**A**) by Pan et al. in [12]; (**B**) by Mehboob et al. in [13]; (**C**) by Indu, S. in [14]; (**D**) our proposed framework.

*4.3. Discussion on Simulations*

The grayscale images, detected and filtered foreground and optical flow of randomly selected frames are shown in Figure 4 for video dataset 1, Figure 6 for video dataset 2, Figure 8 for video dataset 3, Figure 10 for video dataset 4, Figure 12 for video dataset 5 and Figure 14 for video dataset 6, respectively. The filtered foreground images are obtained by adaptive background subtraction by frame differencing, followed by adaptive thresholding and binarization.

Federated optical flow is used to further remove noise from the foreground generated by adaptive background subtraction. Each frame is then assigned a normalized weight based on its relative temporal position using HWHM Gaussian distribution.

Comparisons of activity maps and normalized pixel sensitivity maps derived by Pan et al. in [12], as [A], Mehboob et al. in [13], as [B], Indu, S. in [14], as [C], and our proposed framework, as [D], are shown in Figure 5 for video dataset 1, Figure 7 for video dataset 2, Figure 9 for video dataset 3, Figure 11 for video dataset 4, Figure 13 for video dataset 5 and Figure 15 for video dataset 6, respectively. The approaches presented in [12,13] do not include any temporal relation between past and present frames. Pan et al. in [12] do not present any filter for the pre-processing of the raw images obtained by the sensor. Mehboob et al., in [13], proposed the filtering of the foreground through morphological operations (close and erode), and their approach thus performed better than the approach presented by Pan et al. in [12]. However, both Pan et al. in [12] and Mehboob et al. in [13] failed to showcase the temporal effect of past frames on the present frame, which can be seen in Figure 5 for video dataset 1, Figure 7 for video dataset 2, Figure 9 for video dataset 3, Figure 11 for video dataset 4, Figure 13 for video dataset 5 and Figure 15 for video dataset 6, respectively.

Indu, S., in [14], proposed a spatiotemporal relation between past and present frames; however, the proposed method in [14] failed to provide efficient filtering of the foreground and thus resulted in inaccurate spatiotemporal activity and sensitivity maps due to the noise and undesirable regions detected in the activity map. The proposed framework, in comparison to that of Pan et al. in [12] and that of Mehboob et al. in [13], presents better spatiotemporal activity tracking in terms of normalized pixel sensitivity. Further, the proposed framework outperforms the spatiotemporal activity mapping proposed by Indu, S. in [14] by filtering out the unwanted objects and noise from the detected ROI. The

impact of falsely detected or undesirable objects in the activity map can be derived through multi-object tracking accuracy in the following Section 4.4.

### 4.4. Results

The comparative performance analysis of different approaches in terms of performance parameters is shown in Table 1 for video dataset 1, Table 2 for video dataset 2, Table 3 for video dataset 3, Table 4 for video dataset 4, Table 5 for video dataset 5 and Table 6 for video dataset 6, respectively.

**Table 1.** Comparison of performance parameters by different approaches tested on video dataset 1: [A] by Pan et al. in [12]; [B] by Mehboob et al. in [13]; [C] by Indu, S. in [14]; [D] our proposed approach; [E] the true data obtained using markers.

| Ref. | TPC | TPR (%) | FPC | FPR (%) | TNC | TNR (%) | FNC | FNR (%) | MOTA (%) |
|---|---|---|---|---|---|---|---|---|---|
| **Video Dataset 1 (Traffic Surveillance):** | | | | | | | | | |
| [A] | 67,830 | 92 | 39,114 | 53.09 | 117,558 | 75.03 | 5898 | 3.76 | 43.15 |
| [B] | 66,245 | 89.85 | 32,761 | 44.43 | 123,911 | 79.09 | 7483 | 4.77 | 50.80 |
| [C] | 65,924 | 89.41 | 6273 | 8.51 | 150,399 | 95.99 | 7804 | 4.98 | 86.51 |
| [D] | 65,138 | 88.34 | 1071 | 0.14 | 155,591 | 99.31 | 8590 | 5.48 | 94.38 |
| [E] | 73,728 | 100 | 0 | 0 | 1,56,672 | 100 | 0 | 0 | 100 |

**Table 2.** Comparison of performance parameters by different approaches tested on video dataset 2: [A] by Pan et al. in [12]; [B] by Mehboob et al. in [13]; [C] by Indu, S. in [14]; [D] our proposed approach; [E] the true data obtained using markers.

| Ref. | TPC | TPR (%) | FPC | FPR (%) | TNC | TNR (%) | FNC | FNR (%) | MOTA (%) |
|---|---|---|---|---|---|---|---|---|---|
| **Video Dataset 2 (Traffic Surveillance):** | | | | | | | | | |
| [A] | 83,262 | 94.50 | 43,149 | 48.97 | 99,146 | 69.67 | 4843 | 3.40 | 47.63 |
| [B] | 81,989 | 93.05 | 38,102 | 43.24 | 104,193 | 73.22 | 6116 | 4.29 | 52.46 |
| [C] | 80,594 | 91.47 | 8122 | 9.21 | 134,173 | 94.29 | 7511 | 5.27 | 85.52 |
| [D] | 79,813 | 90.59 | 1622 | 0.18 | 140,673 | 98.86 | 8292 | 5.82 | 94.00 |
| [E] | 88,105 | 100 | 0 | 0 | 142,295 | 100 | 0 | 0 | 100 |

**Table 3.** Comparison of performance parameters by different approaches tested on video dataset 3: [A] by Pan et al. in [12]; [B] by Mehboob et al. in [13]; [C] by Indu, S. in [14]; [D] our proposed approach; [E] the true data obtained using markers.

| Ref. | TPC | TPR (%) | FPC | FPR (%) | TNC | TNR (%) | FNC | FNR (%) | MOTA (%) |
|---|---|---|---|---|---|---|---|---|---|
| **Video Dataset 3 (Traffic Surveillance):** | | | | | | | | | |
| [A] | 106,274 | 96.15 | 41,827 | 37.84 | 78,051 | 65.11 | 4248 | 3.54 | 57.11 |
| [B] | 104,483 | 94.53 | 34,131 | 30.88 | 85,747 | 71.54 | 6039 | 5.03 | 64.09 |
| [C] | 102,173 | 92.44 | 9138 | 8.27 | 110,740 | 92.37 | 8349 | 6.96 | 84.77 |
| [D] | 100,628 | 91.04 | 1122 | 0.10 | 118,756 | 99.06 | 9894 | 8.25 | 91.65 |
| [E] | 110,522 | 100 | 0 | 0 | 119,878 | 100 | 0 | 0 | 100 |

**Table 4.** Comparison of performance parameters by different approaches tested on video dataset 4: [A] by Pan et al. in [12]; [B] by Mehboob et al. in [13]; [C] by Indu, S. in [14]; [D] our proposed approach; [E] the true data obtained using markers.

| Ref. | TPC | TPR (%) | FPC | FPR (%) | TNC | TNR (%) | FNC | FNR (%) | MOTA (%) |
|------|-----|---------|-----|---------|-----|---------|-----|---------|----------|
| **Video Dataset 4 (Sports—Badminton):** | | | | | | | | | |
| [A] | 82,107 | 95.35 | 26,187 | 30.41 | 118,105 | 81.85 | 4001 | 2.77 | 66.82 |
| [B] | 80,926 | 93.98 | 19,223 | 22.32 | 125,069 | 86.68 | 5182 | 3.59 | 74.09 |
| [C] | 78,446 | 91.10 | 5982 | 6.94 | 138,310 | 95.8 | 7662 | 5.31 | 87.75 |
| [D] | 77,102 | 89.54 | 2321 | 2.69 | 141,971 | 98.39 | 9006 | 6.24 | 91.07 |
| [E] | 86,108 | 100 | 0 | 0 | 144,292 | 100 | 0 | 0 | 100 |

**Table 5.** Comparison of performance parameters by different approaches tested on video dataset 5: [A] by Pan et al. in [12]; [B] by Mehboob et al. in [13]; [C] by Indu, S. in [14]; [D] our proposed approach; [E] the true data obtained using markers.

| Ref. | TPC | TPR (%) | FPC | FPR (%) | TNC | TNR (%) | FNC | FNR (%) | MOTA (%) |
|------|-----|---------|-----|---------|-----|---------|-----|---------|----------|
| **Video Dataset 5 (Sports—Sword Fight):** | | | | | | | | | |
| [A] | 66,121 | 91.87 | 23,877 | 33.17 | 128,547 | 81.14 | 5885 | 3.71 | 63.12 |
| [B] | 63,964 | 88.86 | 21,232 | 29.49 | 137,192 | 86.59 | 8012 | 5.05 | 65.46 |
| [C] | 58,372 | 81.10 | 12,121 | 16.84 | 146,303 | 92.35 | 13,604 | 8.58 | 74.58 |
| [D] | 54, 232 | 75.34 | 8962 | 12.45 | 149,462 | 94.32 | 17,744 | 11.2 | 76.35 |
| [E] | 71,976 | 100 | 0 | 0 | 158,424 | 100 | 0 | 0 | 100 |

**Table 6.** Comparison of performance parameters by different approaches tested on video dataset 6: [A] by Pan et al. in [12]; [B] by Mehboob et al. in [13]; [C] by Indu, S. in [14]; [D] our proposed approach; [E] the true data obtained using markers.

| Ref. | TPC | TPR (%) | FPC | FPR (%) | TNC | TNR (%) | FNC | FNR (%) | MOTA (%) |
|------|-----|---------|-----|---------|-----|---------|-----|---------|----------|
| **Video Dataset 6 (Sports—Tennis):** | | | | | | | | | |
| [A] | 46,185 | 94.34 | 20,863 | 42.61 | 160,602 | 88.50 | 2768 | 1.52 | 55.87 |
| [B] | 43,266 | 88.38 | 18,286 | 37.35 | 163,179 | 89.92 | 5687 | 3.13 | 59.52 |
| [C] | 38,128 | 77.89 | 12,112 | 24.74 | 169,353 | 93.32 | 10,825 | 5.97 | 69.29 |
| [D] | 37,109 | 75.81 | 8934 | 18.25 | 172,531 | 95.08 | 11,844 | 6.52 | 75.23 |
| [E] | 48,953 | 100 | 0 | 0 | 181,465 | 100 | 0 | 0 | 100 |

The average performance of the proposed framework for traffic surveillance has been obtained and compared with contemporary traffic surveillance multi-object tracking systems [28,29] (MOTA average (%) calculated for the first ten video sequences of [28,29]). Further, the average performance of the proposed framework for sports analytics has been obtained and compared with a contemporary sports activity tracking system [30] (MOTA average (%) obtained from the RGB sequence of [30]). The comparison of the performance of the proposed framework with that of [28–30] in terms of average MOTA (%) is shown in Figure 16.

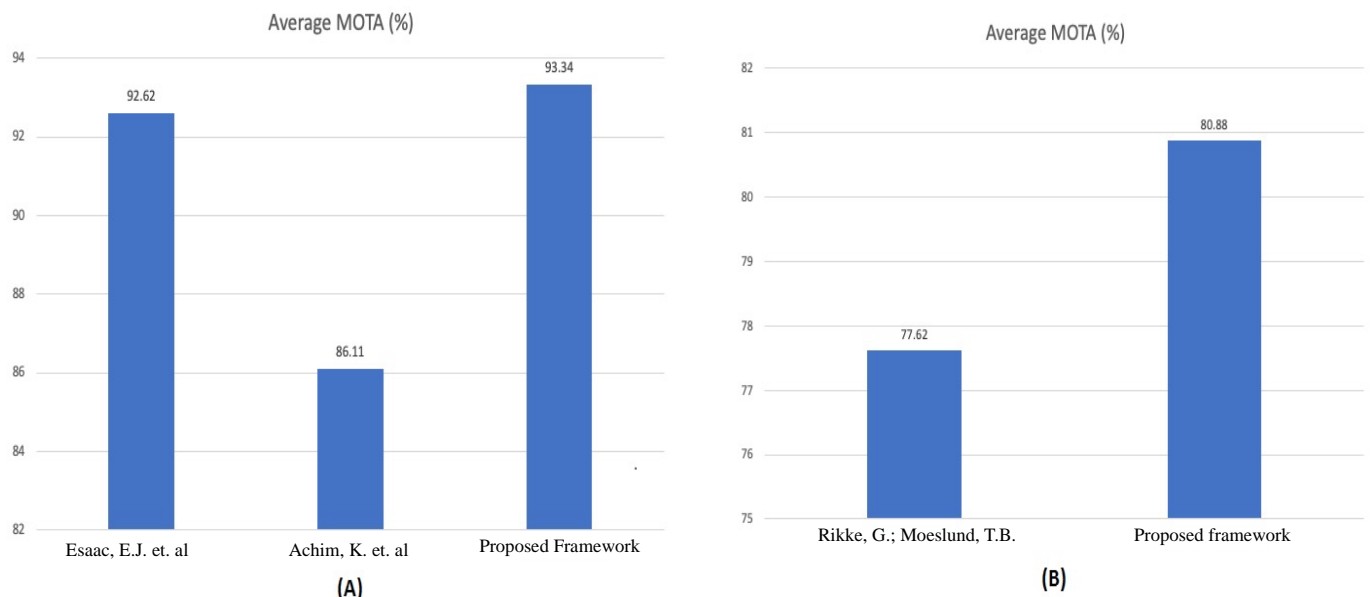

**Figure 16.** Comparison of the performance in terms of average MOTA %: (**A**): Comparison of the average MOTA % of the proposed framework with that of Isaac, E.J. et. Al in [28] and Achim, K. et.al in [29]; (**B**): Comparison of the average MOTA % of the proposed framework with that of Rikke, G.; Moeslund, T.B. in [30].

The average MOTA (%) of the proposed framework for traffic surveillance is obtained as 93.34%, as calculated by Equation (8):

$$\{MOTA\ (dataset\text{-}1) + MOTA\ (dataset\text{-}2) + MOTA\ (dataset\text{-}3)\}/3; \tag{8}$$

The average MOTA (%) of the proposed framework for sports analytics is obtained as 80.88%, as calculated by Equation (9):

$$\{MOTA\ (dataset\text{-}4) + MOTA\ (dataset\text{-}5) + MOTA\ (dataset\text{-}6)\}/3; \tag{9}$$

## 5. Conclusions

The accuracy of the data collected by sensors and the spatiotemporal understanding of the scene are two key components of a successful computer vision system. Most of the advanced computer vision systems address both of the abovementioned components using a highly complex computation model, which may be a problem for most of the systems with limited resources. Through this article, a framework for spatiotemporal activity mapping capable of handling the trade-off between resource limitations and the performance of the computer vision system is proposed.

The framework evaluates the scene spatiotemporally and produces adaptive activity maps for the re-configuration of the sensor such that the region(s) of importance can be captured in the center of the sensor's field of view. The framework utilizes simple image-processing tools such as adaptive background subtraction, binarization, thresholding and federated optical flow for pre-processing the sensor data. Half-width Gaussian distribution is used for the temporal relationship between the present and past frames. The simple model of the proposed framework results in a low computational complexity and thus low resource utilization.

The performance of the framework is compared in terms of multi-object tracking accuracy (MOTA) and has been tested on multiple traffic surveillance and sports datasets. The framework outperforms the contemporary systems presented in [12–14,28–30]. The framework showcased a 0.71% better average MOTA compared to [28] and an 8.39% better average MOTA compared to [29] when tested on traffic surveillance datasets (i.e., datasets 1, 2 and 3). The framework further showcased a 4.21% better average MOTA compared

to [30] when tested on sports datasets (i.e., datasets 4, 5 and 6). Artificial intelligence (AI)-based systems [16–19] require high computation and storage capabilities to train the model and further require iteratively changing the training model in unforeseen conditions, illumination changes, etc. AI-based multi-object tracking systems are also susceptible to adversarial attacks, which makes the timely training of the system critically necessary. The proposed framework performs similarly to the AI-based multi-object detection systems proposed in [16–19] without the high computation or storage requirements needed to handle unforeseen conditions. Thus, the proposed framework showcases a balance in terms of resource utilization and performance.

**Author Contributions:** S.: Lead author of the manuscript (corresponding author), conceptualization and methodology, writing—original draft preparation, investigation and editing; I.S.: Second author, research design, guidance and reviewing. All authors have read and agreed to the published version of the manuscript.

**Funding:** This research received no external funding.

**Data Availability Statement:** The proposed framework has been tested on three surveillance datasets (i.e., video dataset 1, video dataset 2 and video dataset 3) and three sports datasets (video dataset 4, video dataset 5 and video dataset 6). The video datasets and codes for the simulation models for the proposed framework can be accessed by clicking on this https://drive.google.com/drive/folders/1a3SG5qEOL25e7pFIPOb0GCvGKL6X7ghN (last accessed on 12 December 2022).

**Acknowledgments:** This work was carried out under the supervision of Indu Sreedevi at the Department of ECE, Delhi Technological University, New Delhi, India, and Shashank expresses immense gratitude to his guide and to UGC for enlightening him throughout the process.

**Conflicts of Interest:** The authors declare that they have no conflict of interest. The authors declare that they have no known competing financial interests or personal relationships that could have appeared to influence the work reported in this article.

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
