# Peer review of "Spatiotemporal Activity Mapping for Enhanced Multi-Object Detection with Reduced Resource Utilization"

_electronics, doi:10.3390/electronics12010037_

Round 1

Reviewer 1 Report

This paper presents a framework to reconfigure computer vision systems for better object tracking through spatiotemporal evaluation of the scene. The proposed framework relies on a model-based approach with low computation complexity. Here are the comments:

Sections 1-2:

Line 35: "Some robotic systems...": Not sure what this sentence means?

Line 41: "Accuracy of data": accuracy in general describes the accuracy of the 3D information related to a feature. I don't think this term is accurate here. 

Line 44: How are sensor calibration and data processing related? I am not sure what is the logic of this sentence.

Line 92: You described the camera pair configuration, how about the single camera case?

Line 102: Again, "accurate" here is not precise.

Line 109: Can you make it clear what is the definition of "present and past events"? How long is the duration of present/past events?

Line 145: For deep learning based approach, if the model is pre-trained, directly applying the model should not be very time-consuming. Can you clarify this?

Section 3

Line 152: Heading needs to be more informative.

Line 158: In Line 92, you mention that only one camera will be used to generate an activity map, but here you are saying generating for each sensor.

Line 158: You need to briefly introduce this framework at a high level. Leaving such a framework without explanation is not appropriate.

At the end of the introduction of Section 3(Line 163), include a brief introduction of what will be covered in each of the following subsections. It will be easy for readers to follow.

Subsection "Adaptive background subtraction": 

"For example... The background information... Then..." What is the logic? What is the meaning of equation 1? I don't see "adaptive background subtraction" explained.

Subsection "Normalized Halfwidth Gaussian Distribution": 

Equation 2: What is x? What are mu and alpha, you mentioned they are mean and std, but of what? The figure is not necessary, everyone knows the Gaussian distribution. The problem is that you didn't provide enough information related to how to use this in your application.

Subsection "Federated Optical flow": the details related to the approach are not enough.

Line 233: How are binarization and thresholding conducted? What is the purpose? Are you sure it is binarization followed by thresholding?

I don't see any correspondence between the text and Figure 1 (e.g., where is S1i, S2i ... H(t-i), X, N, R, C, Y)

Major problems of Section 3: 

What is your contribution and what is the novelty of this approach? For your contribution/novelty, your write-up should be in more detail.

Section 4:

Line 288-303: I feel like this should be introduced in the method (Section 3)

Figure 4 (c): The optical flow is not clear, same for the following figures.

Equations 9 & 10: are they necessary?

Major problems of Section 4:

I don't see any analysis related to Figures 4 - 11. You have 8 figures without any discussion. From these figures, qualitatively, I don't see major differences between the [c] by Indu, s. and [d] your approach. Even for results from approaches [a] and [b], they successfully identified the ROI. Although you showed the quantitative results indicating that the proposed approach achieved the best results pixel-wise, does your approach make a difference for the application of configuring the systems to focus on the ROI. Can you show some samples that the results from [a][b][c] fail to configure the system to the ROI but your approach can?

Also, you mention that your approach is with low computation complexity, I don't see any results/discussions related to this. You have to prove your statement.

Minor comments

Line 63: result-> results

Line 93: is be used -> is used

Line 123: "used centroid detection using" avoid repetition 

Line 187: "For example..." this is not a complete sentence

Figure 4 "Results post binarization" -> resulted

Reviewer 2 Report

The paper is easy to understand. There are some mistakes. 

1. Eq.3 seems to be wrong. The alpha should not be in the square.  

2. Where is the background subtraction  B in Figure1?

When do you use the B? 

3.  Some figures are blurred. The author may consider to upload high-quality figures. 

4. The complexity analysis of the proposed method should be provided, such as the FLOPs and the number of parameters.

5.Is it possible to apply to the vehicle Reid datasets, like VeRi-776 [a] and VehicleNet [b]? It would be great to have a discussion.

[a] Xinchen Liu, Wu Liu, Huadong Ma, Huiyuan Fu: Large-scale vehicle re-identification in urban surveillance videos. ICME 2016

[b] Zhedong Zheng, et al. VehicleNet: Learning Robust Visual Representation for Vehicle Re-identification. IEEE Trans. Multimedia (2020). 

Round 2

Reviewer 1 Report

The majority of comments have been resolved. However, the quality of the figures needs to be improved.